# STAT3 Inhibitors: A Novel Insight for Anticancer Therapy of Pancreatic Cancer

**DOI:** 10.3390/biom12101450

**Published:** 2022-10-09

**Authors:** Xin Li, Wenkai Jiang, Shi Dong, Wancheng Li, Weixiong Zhu, Wence Zhou

**Affiliations:** 1The First Clinical Medical College, Lanzhou University, Lanzhou 730000, China; 2Department of General Surgery, The Second Hospital of Lanzhou University, Lanzhou 730030, China

**Keywords:** STAT3, pancreatic cancer, drug therapy, molecular targeted therapy

## Abstract

The signal transducer and activator of transcription (STAT) is a family of intracellular cytoplasmic transcription factors involved in many biological functions in mammalian signal transduction. Among them, STAT3 is involved in cell proliferation, differentiation, apoptosis, and inflammatory responses. Despite the advances in the treatment of pancreatic cancer in the past decade, the prognosis for patients with pancreatic cancer remains poor. STAT3 has been shown to play a pro-cancer role in a variety of cancers, and inhibitors of STAT3 are used in pre-clinical and clinical studies. We reviewed the relationship between STAT3 and pancreatic cancer and the latest results on the use of STAT3 inhibitors in pancreatic cancer, with the aim of providing insights and ideas around STAT3 inhibitors for a new generation of chemotherapeutic modalities for pancreatic cancer.

## 1. Introduction

The signal transducer and activator of transcription (STAT) is a family of intracellular cytoplasmic transcription factors involved in many biological functions in mammalian signal transduction [1]. Among them, STAT3 is involved in cell proliferation, differentiation, apoptosis, and inflammatory responses, and also STAT3-related signaling pathways are aberrant over-activated in many types of cancer and are strongly associated with poor patient prognosis [2,3]. In addition, excessive activation of STAT3 within tumor cells and other cells in the tumor microenvironment (TME) mediates a series of extracellular signals that enhances the immune inflammatory response in the TME, driving tumor cell proliferation, invasion, and metastasis, while strongly suppressing the anti-tumor immune response and creating an immunosuppressive microenvironment [1,4].

Despite the advances in the treatment of pancreatic cancer in the past decade, the prognosis for patients with pancreatic cancer remains poor. The number of cases and deaths caused by pancreatic cancer has more than doubled from 1990 to 2017 and the morbidity is likely to continue to rise as the population ages [5]. The main clinical issue that needs to be addressed for patients with pancreatic cancer is the search for more effective treatments and drug targets to prolong the survival of patients with pancreatic cancer. STAT3 has been shown to play a pro-cancer role in a variety of cancers and inhibitors of STAT3 are used in pre-clinical and clinical studies. In this paper, we reviewed the relationship between STAT3 and pancreatic cancer and the latest results on the use of STAT3 inhibitors in pancreatic cancer, with the aim of providing insights and ideas around STAT3 inhibitors for a new generation of chemotherapeutic modalities for pancreatic cancer.

## 2. STAT3 and Its Pathways

### 2.1. Structure of STAT3

The STAT family consists of seven members, namely STAT1, STAT2, STAT3, STAT4, STAT5A, STAT5B, and STAT6 [6]. STAT3 has two splicing isoforms with different functions, STAT3α and STAT3β. The domain of STAT3α is intact, but STAT3β lacks the 55 C-terminal amino acid residues, which has the function of negatively regulating transcription. STAT3β has better specific DNA binding activity than STAT3α [7,8]. STAT3 acts mainly to promote tumor growth and immunosuppression [9]. The gene encoding STAT3 is located on human chromosome 17 and the protein consists of 770 amino acids, of which tyrosine at position 705 and serine at position 727 can be phosphorylated for functional activation [10]. Similar to other STAT families, STAT3 protein contains six structural domains, namely N-terminal domain (NTD), coiled-coil domain (CCD), DNA-binding domain (DBD), linking domain (LD), SH2 domain, and transcriptional activation domain (TAD), which have different functions [11]. The NTD promotes the formation of STAT dimers, enabling their subsequent binding with transcription factors [11]. The coiled-coil domain of STAT3 is essential for its SH2 domain-mediated receptor binding and subsequent activation induced by epidermal growth factor and interleukin-6 [12]. DBD can recognize and bind DNA sequences in regulatory regions of target genes [11]. LD connects DBD to the SH2 domain [13]. SH2 is the most highly conserved STAT domain and plays a key role in signaling by binding to specific phosphorylated tyrosine motifs [14]. The linear structure and three-dimensional structure of STAT3 are shown in Figure 1.

STAT3 in its inactive state is located in the cytoplasm and when STAT3 is phosphorylated, its SH-2 mediates STAT3 to produce a dimer and STAT3 translocates into the nucleus to transmit extracellular signals into the nucleus [9]. In the nucleus, STAT3 interacts with the corresponding target gene, leading to transcription and expression of the target gene, playing a key role in a variety of biological events [15].

### 2.2. Signal Pathway of STAT3

#### 2.2.1. Classic Signaling

The JAK/STAT pathway is one of the most important signaling pathways regulating cellular function, and dozens of cytokines and growth factors have been identified in this pathway, including interleukins (IL), interferons, and angiogenic factors [11]. IL-6 is an inflammatory mediator that was initially thought to stimulate antibody production by B cells [16]. Since chronic inflammation can promote tumor progression, high expression of IL-6 can also be present in TME. In TME, IL-6 can be produced by a variety of cells, including macrophages, neutrophils, fibroblasts, and tumor cells themselves [17]. IL-6 can directly cause cancer cell proliferation, can also promote the production of other inflammatory factors in TME, and recruit large numbers of immune cells [18]. Therefore, IL-6 is the key substance in chronic inflammation and tumor progression. In response to IL-6 stimulation, the JAK/STAT3 pathway is phosphorylated, forming the critical IL-6/JAK/STAT3 pathway in the human body, which is involved in the processes of rheumatoid arthritis, inflammatory bowel disease, and many human cancers [9,19,20].

At present, four key JAK molecules have been identified, namely JAK1, JAK2, JAK3, and TK2 [21]. Among them, JAK3 is only expressed in cells of the hematopoietic and lymphoid systems, while the remaining three are expressed in almost all cells [11,22,23]. JAK contains seven homology domains (JH1–7) and forms four structural domains: JH1, JH2, SH2, and FERM [23,24].

The classical IL-6 signaling pathway is mediated by IL-6 with the membrane-bound receptors IL-6 receptor α (IL-6Rα) and gp130, resulting in an IL-6/IL-6R/gp130 complex that leads to intracellular JAK activation, including JAK1, JAK2, and TYK2 [25]. JAK protein binds to the intracellular structural domain of gp130, leading to phosphorylation of the tyrosine residue of gp130, forming a binding site for STAT3. After STAT3 recognition and binding to the phosphotyrosine docking site, the attached JAK enzyme with activity phosphorylates STAT3 at position 705 tyrosine, and the phosphorylated STAT3 forms a dimer that enters the nucleus and binds its downstream target genes [26]. The classical IL-6/JAK/STAT3 signaling pathway must first achieve the formation of a binary complex between IL-6 and IL-6R, and only then can it bind to gp130. The JAK/STAT3 pathway is primarily negatively regulated by cytokine signaling inhibitors, tyrosine phosphatases, and STAT3 inhibitors [27]. The classical IL-6 signaling pathway is shown in Figure 2.

#### 2.2.2. Trans-Signaling

In addition, there is also IL-6 trans-signaling (Figure 2), which is mediated through the soluble form of IL-6Rα (sIL-6Rα) and enhances the IL-6 pathway in cells lacking IL-6R [28]. The main mode of production of sIL-6Rα is selective splicing of precursor mRNA or cleavage of membrane bound IL-6R by a disintegrin and metalloprotease 10 (ADAM10) and ADAM17 [29]. When IL-6 binds to sIL-6R, the complex is able to bind and induce dimerization of gp130, thereby activating downstream signaling pathways. IL-6 trans-signaling is mainly negatively regulated by soluble gp130 [30].

#### 2.2.3. Other Signal Pathway

More and more evidence showed that non-coding RNAs are involved in the regulation of STAT3. These regulations include direct targeting of STAT3 and regulation of molecules upstream of STAT3. For example, miR-124 and miR-181d can directly bind to STAT3 and inhibit its expression, while miR-133a, mi-451a, and miR-218 can indirectly inhibit STAT3 activation by targeting STAT3 upstream molecules [31,32,33,34,35]. In addition, there are many miRNAs that promote STAT3 expression, including miR-221, miR-222, miR-30b, and miR-147 [36,37,38]. Similarly, lncRNAs can also regulate STAT3 expression and activation directly or indirectly through a variety of mechanisms, and this regulation is mostly achieved through endosomal competitive RNAs. For example, LINC00115 and CASC9 can further regulate STAT3 function by regulating downstream miRNA in cancer disease [39,40]. Moreover, activation of STAT3 can be induced by phosphorylation of some receptor tyrosine kinases themselves. For example, activation of fibroblast growth factor receptor-1 can induce SRC-mediated STAT3 activation, and increased expression of the epidermal growth factor receptor can also activate STAT3 signaling, thereby promoting the aerobic glycolytic pathway in cells [41,42].

### 2.3. STAT3 in Immune Cells, Stromal Cells, and Epithelial Cells

TME consists of tumor cells, mesenchymal cells, and immune cells, which play an important role in the development of tumors [43]. As a key molecule in the oncogenic mechanism, STAT3 plays a role in the TME [44]. T cells play a central role in cancer immunity. Overactivation of STAT3 in immune cells can inhibit the aggregation of T-cell subsets [1]. For example, IL-35-promoted STAT3 activation, reduces the infiltration and function of CD8+ T cells in pancreatic cancer by inhibiting the expression of CXCR3, CCR5, and IFN-γ [45]. Ablation of STAT3 in CD8+ T cells increased their anti-cancer activity [46].

Overexpression of STAT3 in macrophages can also inhibit the antitumor immune function of T cells. STAT3 can directly induce CD163 expression in macrophages [47]. When STAT3 activity is increased in macrophages, the immune response process of T cells is impaired [48].

Previous proteomic results have confirmed that the activation of tumor-associated fibroblasts (CAF) in pancreatic cancer is regulated by STAT3 [49]. STAT3 activation in CAFs results in the production of several immunosuppressive factors, such as IL-6, TGF-β, and CCL [1]. Pancreatic stellate cells (PSC) are indispensable fibroblasts in pancreatic cancer TME. The cytokines produced by PSC can enhance the phosphorylation of STAT3 and the differentiation of myeloid-derived suppressor cells, and promote the immunosuppressive microenvironment [50].

STAT3 activation in epithelial cells can also contribute to cancer progression. For instance, IL-11-mediated STAT3 activation in gastric cancer epithelial cells results in melanoma 2 up-regulation, which promotes epithelial cell migration and development [51].

## 3. STAT3 in Pancreatic Cancer

Pancreatic cancer is the leading cause of cancer deaths worldwide. In 2017, there were 448,000 cases of pancreatic cancer occurring worldwide, with a 2.3-fold increase in deaths from 196,000 in 1990 to 4.41 million in 2017, and a 2.1-fold increase in disability-adjusted life years due to pancreatic cancer [5]. Not only that, but the incidence of PDAC is increasing by 0.5–1.0% per year and is expected to be the second leading cause of cancer-related mortality by 2030 [52]. As a key molecule in the mechanism of carcinogenesis, STAT3 plays a role in tumor cells and other cells in the TME of pancreatic cancer [4]. On the one hand, phosphorylated STAT3 regulates the expression of its downstream target genes and directly promotes the proliferation, invasion, and apoptosis of cancer cells; on the other hand, STAT3 can induce overexpression of a variety of cytokines and chemokines associated with cancer progression in TME, thus acting as a link between tumor and inflammation and tumor and immune cells [2,4]. In this section, we mainly introduce the role of STAT3 in pancreatic cancer.

### 3.1. Tumor Progression and STAT3

STAT3 pathway plays a role in pancreatic cancer, in which IL-6 is the driving factor. IL-6 enhances STAT3 phosphorylation in pancreatic cancer PANC-1 cell line [53]. Vascular endothelial growth factor (VEGF), one of the target genes downstream of STAT3, was significantly elevated in tumor tissues and tissues with lymph node metastases [54]. The high expression of STAT3 in pancreatic cancer is positively correlated with the expression of VEGF and VEGF-C, indicating that STAT3 activation may be associated with angiogenesis in pancreatic cancer [55].

The upregulation of STAT3 in pancreatic cancer may also depend on the expression and function of other genes. Ubiquitin-specific peptidase 5 (USP5) correlates with serum CEA and CA19-9 levels in pancreatic cancer patients, and high USP5 expression is an unfavorable prognostic factor for pancreatic cancer. Research DirectionsUSP5 enhances STAT3 signaling in cancer cells and promotes migration and invasion of pancreatic cancer [56]. LncRNA FGD5-AS1 in pancreatic cancer can pass through exosomes to TME, where FGD5-AS1 interacts with p300, leading to STAT3 acetylation, which promotes NF-κB nuclear translocation and transcriptional activity and stimulates macrophage polarization toward the M2 type, thereby enhancing pancreatic cancer cell proliferation [57].

### 3.2. Metastasis and STAT3

Metastatic lesions are the main cause of death in cancer patients. Metastasis of pancreatic cancer cells can allow tumors to grow in distant organs and evade immune surveillance [58]. Autologous amplified Notch signaling in pancreatic cancer promotes macrophage recruitment and differentiation to the M2 phenotype through activation of IL-8 and CCL2 paracrine signaling downstream. In turn, M2 macrophages secrete IL-6 to activate STAT3 and directly inhibit miR-124 through the conserved STAT3 binding site in its promoter, thus promoting invasion and metastasis [59]. In contrast, some miRNAs are carcinogenic. MiR-301a is highly expressed in human pancreatic cancer patients, which can target SOCS5 to activate JAK/STAT3 and promote the migration and invasion of pancreatic cancer cells [60].

Epithelial-mesenchymal transition occurs during the development and metastasis of various primary tumors [61]. The EMT mechanism is responsible for the metastasis of tumor cells and their spread to various organs and tissues of the body [62]. IL-6 overexpression is manifested by aberrant activation of STAT3 in tumor tissue, leading to decreased E-cadherin and increased vimentin protein expression [63]. Gp130/STAT3 pathway regulates the TGF-β expression in cancer stem cells such as cells in pancreatic cancer, thereby enhancing the function of TGF-β/Smad pathway in EMT [64]. Similarly, STAT3 signaling pathway activation induced by S100A16 also promoted the migration and EMT of pancreatic cancer cells [65].

### 3.3. Immunosuppression and STAT3

Immune escape and immunosuppression are among the features in TME; STAT3 in cancer cells can inhibit the activation of dendritic cells and promote enhanced metabolism of substances in the TME [66]. Immunosuppression in pancreatic cancer TME is manifested by a lack of CD8+ T-cell numbers and decreased function [67]. The number of CD8+ T cells in pancreatic cancer TME is significantly reduced when IL-35-producing regulatory T cells and B cells aggregate, due to IL-35-promoted activation of STAT3 inhibiting the intratumoral infiltration and effector functions of CD8+ T cells by suppressing CXCR3, CCR5, and IFN-γ expression [45]. Moreover, the polyadenosine diphosphate ribose polymerase inhibitor pamiparib upregulated programmed death ligand 1 (PD-L1) expression on the surface of pancreatic cancer cells in vitro and in vivo via the JAK2/STAT3 pathway, with a significant increase in CD8+ T cells in TME [68]. PSC produces cytokines that enhance STAT3 phosphorylation and myeloid-derived suppressor cell (MDSC) differentiation and promote an immunosuppressive microenvironment [50].

STAT3 can be considered as the intersection of multiple oncogenic signaling pathways, activated in immune cells and cancer cells in pancreatic cancer TME, promoting the production of immunosuppressive factors that alter gene expression programs and suppress anti-tumor immune responses. Therefore, STAT3 plays a role in pancreatic cancer and its TME, leading to tumor-induced immunosuppression.

### 3.4. STAT3 and Drug Resistance

Although the treatment of pancreatic cancer has made progress in recent years, the drug resistance of pancreatic cancer is still one of the important factors affecting the prognosis of pancreatic cancer patients. For many years, gemcitabine-based chemotherapy regimens have been the standard of care for advanced pancreatic cancer, yet its resistance is one of the reasons for the low overall survival and susceptibility to recurrence of pancreatic cancer patients [69].

MiR-1266 overexpression is associated with poor prognosis in patients with pancreatic cancer. In pancreatic cancer cells BxPC-3 and SW1990, miR-1266 significantly increased the nuclear aggregation of STAT3 and NF-κB/p65 and the phosphorylation level of STAT3 and NF-κB/p65 in the nucleus, activated the expression of its downstream genes Bcl2, Bcl-xL, MCL1, and BIRC5, exerted the anti-apoptotic ability of pancreatic cancer cells, and enhanced resistance to gemcitabine [70].

Radiotherapy enhances the Warburg effect in pancreatic cancer cells, resulting in a sustained increase in lactate secretion; but lactate can lead to significant up-regulation of phosphorylated STAT3 in MDSCs via the G protein-coupled receptor 81/hypoxia-inducible factor-1α pathway, and MDSCs are activated, promoting the development of an immunosuppressive microenvironment in pancreatic cancer, leading to resistance to radiotherapy [71].

## 4. STAT3 Inhibitors and Cancer

Based on its important biological function in cancer, STAT3 has been investigated as a partial inhibitor for the next clinical therapeutic target in oncology. However, up to now, most of the inhibitors applied to STAT3 targets have been designed to indirectly inhibit the biological function of STAT3 by blocking its upstream signaling mechanism, including IL-6 inhibitors, JAK inhibitors, and inhibitors of various growth factor receptors. In addition, the importance of the SH2 structural domain in STAT3 has been increasingly recognized in recent years and a series of small molecule inhibitors has been designed to directly target STAT3, the functions of which have been demonstrated in preclinical studies of various cancers. Among them, some natural compounds have also shown anti-tumor activity and have demonstrated inhibition of STAT3 in vivo and in vitro experiments. STAT3 target-related inhibitors of pancreatic cancer are summarized in Table 1.

### 4.1. Direct Targeting of STAT3 Inhibitors

Post-transcriptional inhibition of STAT3 synthesis in cancer cells using small interfering RNA (siRNA) is the most common method of inhibiting the STAT3 pathway. In vitro, STAT3 inhibitor STX-0119 showed cytotoxicity to a variety of pancreatic cancer cell lines, which showed weak PD-L1 expression [72]. It is worth noting that siRNA, despite its disadvantages of poor stability and low cellular uptake, is protected by exosomes for delivery and transport to help it work, and can even be further enhanced in its oncogenic effects by a co-delivery system formed by nanoparticles and conventional chemotherapeutic drugs [73,74,75].

Peptide inhibitors can be designed by amino acid residues in the STAT3 structure, so as to target specific domains. But peptide inhibitors are susceptible to degradation in vivo and have difficulty passing through cell membranes. Therefore, non-peptides that inhibit STAT3 phosphorylation are currently the ideal STAT3 inhibitors. Napabucasin, a class of tumor stem cell inhibitors targeting STAT3, is currently in a multicenter phase 3 clinical trial to test its anti-cancer effects [76]. The development of small molecule STAT3 inhibitors is mainly based on the design of structures and the screening of compound libraries. Current small molecule STAT3 inhibitors primarily target the SH2 structural domain, thereby inhibiting the formation of activated STAT3 dimers. N4 is a newly designed small molecule STAT3 inhibitor that binds directly to the STAT3 SH2 structural domain, thereby effectively inhibiting the binding of STAT3 dimers, STAT3-EGFR and STAT3-NF-kB. In animal models of pancreatic cancer, N4 was also well tolerated, inhibited tumor growth and metastasis, and significantly prolonged survival in tumor-bearing mice [77]. C188-9 is also a novel and potent STAT3 inhibitor targeting the SH2 structural domain and was first reported to inhibit granulocyte colony-stimulating factor-induced STAT3 phosphorylation and promote apoptosis in acute myeloid leukemia cell lines [78]. In pancreatic cancer, C188-9 combined with decitabine treatment modulates DNA methyltransferase 1 and significantly reverses the hypermethylation status of tumor suppressor genes in pancreatic cancer [79]. Through bioactivity screening and molecular docking, the investigators identified L61H46, a small molecule targeting STAT3 signaling in vitro and in vivo, which effectively reduced pancreatic cancer cell viability, significantly inhibited STAT3 phosphorylation and subsequent nuclear translocation, and showed no adverse effects on the liver, heart, and kidney cells in a xenograft tumor model [80]. The discovery and study of this novel anti-cancer drug that directly targets STAT3 may offer potential clinical benefits for the treatment of pancreatic cancer.

At present, some natural compounds exhibit anticancer activity, which kills cancer cells and has less effect on normal tissue [81]. Shikonin is a novel anticancer drug isolated from Lithospermum that promotes autophagy in cancer cells [82]. In pancreatic cancer, the expression of cleaved-caspase3 was elevated in Shikonin-treated PANC-1 and BxPC3 cells, and Shikonin inhibited immune escape from pancreatic cancer by inhibiting the NF-kB/STAT3 pathway and promoting the degradation of PD-L1 [83]. Triptomycin A (TA) is a secondary metabolite produced by Actinomyces, which binds directly to STAT3 and inhibits amino acid phosphorylation at position 705 of STAT3. In vitro TA also inhibits the proliferation, migration, and invasion of pancreatic cancer cell lines. In vivo TA also significantly blocked the growth of pancreatic tumors without significant toxicity [84]. In addition, some natural inhibitors of STAT3 can enhance the sensitivity of pancreatic cancer cells to the drug, for example, Rhein and Alantolactone enhance tumor sensitivity to erlotinib in a xenograft model and Berbamine induces apoptosis in pancreatic cancer cells in vitro in concert with gefitinib [85,86,87].

AZD9150, an antisense oligonucleotide containing a restricted ethyl modification targeting STAT3, reduced STAT3 expression in a wide range of preclinical cancer models and showed antitumor activity in lymphoma, neuroblastoma, and lung cancer models [88,89]. In phase 1 clinical trial that included 30 patients, AZD9150 was well tolerated and efficacious in heavily pretreated patients with diffused large B-cell lymphoma [90]. In pancreatic cancer, the clinical effectiveness and safety of AZD9150 is being evaluated (NCT02983578). In addition, there are several small-molecule inhibitors currently in clinical trials for solid tumors, such as OPB-31121 in hepatocellular carcinoma (NCT01406574), and BBI608 (Napabucasin) in colorectal cancer (NCT01830621). Although none of the STAT3 inhibitors developed to date have been included as clinical candidates for pancreatic cancer and further work is needed to achieve this, the above studies demonstrate that direct targeting of STAT3 is feasible.

### 4.2. Indirect Targeting of STAT3 Inhibitors

#### 4.2.1. Inhibitors of IL-6 and IL-6R

Upstream cytokines of STAT3 and their receptors are essential for the activation of STAT3 function. Recombinant cytokines or cytokine/receptor blockers have been used in several studies [91,92]. Because the IL-6 classical signaling pathway must be achieved through the binding of IL-6 to IL-6R and gp130, the main methods currently used to inhibit IL-6-mediated signaling are to block the receptor subunit of IL-6 or to neutralize IL-6 directly.

Tocilizumab is a highly specific monoclonal antibody that recognizes IL-6R [93]. Serum levels of IL-6 and soluble IL-6R are elevated in RA patients after treatment with tocilizumab, which is currently approved by the FDA for the treatment of RA and juvenile idiopathic arthritis [94,95]. The effects of tocilizumab in cancer disease are similar to those of RA, with patients receiving tocilizumab experiencing functional IL-6R blockade, increased serum IL-6 and soluble IL-6R levels, decreased levels of phosphorylated STAT3, and more active T-cell function in vivo, secreting more interferon-gamma and tumor necrosis factor-alpha [96].

Tocilizumab has been shown to exert anticancer effects in pancreatic cancer in ex vivo experiments. In a mouse transplantation tumor model, tocilizumab blockade of IL-6 significantly abrogated mesenchymal stromal cell-mediated tumor promotion and delayed tumor formation [97], and inhibition of IL-6 signaling by tocilizumab partially reversed the EMT phenotype of pancreatic ductal epithelial Kras mutant cells [98]. PSCs and cancer cells in pancreatic cancer TME release ATP, which activates the P2X7 receptor, promoting PSC proliferation, collagen secretion, and IL-6 secretion, and P2X7R-conditioned cultures also stimulate PSCs to activate the JAK/STAT3 signaling pathway in cancer cells, leading to pancreatic cancer cell migration, which is inhibited by tocilizumab [99].

Currently, siltuximab is the only FDA-approved drug for the treatment of idiopathic multicentric Castleman disease (iMCD) and is approved for use in all patients with HHV-8 and HIV-negative multicentric Castleman disease [100]. Siltuximab can fully eliminate IL-6 signal transduction in vivo and in vitro, and inhibit bile duct cancer, oral cancer, and lung cancer cells [101,102,103]. In a phase I/II clinical trial evaluating solid tumors including pancreatic cancer, siltuximab was well tolerated by patients, but no clinical activity was observed and the incidence of adverse events such as abnormal liver function and fatigue was >10% [104]. Therefore, although the anti-cancer effect of siltuximab is promising in in vivo and in vitro experiments, and further research is needed to improve the prognosis of pancreatic cancer patients in clinical setting.

Bazedoxifene is a third-generation selective estrogen receptor modulator [105]. Bazedoxifene inhibits IL-6/IL-6R/GP130 complex, thereby interfering with downstream STAT3 activation. In preclinical studies of pancreatic cancer, bazedoxifene has shown promising antitumor activity, such as inhibiting cancer cell growth and migration, and can synergistically enhance the antitumor effect of other drugs [106,107]. Bazedoxifene also showed no apparent toxicity in vivo [108]. Clinical trial of the effect of bazedoxifene in patients with metastatic pancreatic adenocarcinoma is ongoing (NCT04812808).

#### 4.2.2. Inhibitors of JAKs

Various JAK inhibitors are undergoing preclinical and clinical studies and tofacitinib and baricitinib are two oral JAK inhibitors that have been approved for use and are currently available for the treatment of autoimmune diseases including RA [109,110]. Tofacitinib was the first JAK inhibitor studied, inhibiting mainly JAK1 and JAK3. In cancer cells, tofacitinib blocks the differentiation of NK cells to a resistant phenotype, thus overcoming tumor immune escape [111]. Baricitinib inhibits cytokines and chemokines that are essential for immune cell recruitment [112]. This means that inhibition of JAK may eliminate the inflammatory response in TME, thereby blocking and attenuating the pro-cancer effect of immune cells on pancreatic cancer.

Ruxolitinib also belongs to the first generation of JAK inhibitors and is a competitive compound for ATP [113]. In vivo, ruxolitinib selectively inhibits the activation of JAK and STAT3, increases cytotoxic T lymphocyte infiltration in TME and induces a Tc1/Th1 immune response [114]. Vesicular stomatitis virus (VSV) is an oncolytic virus and combining VSV with ruxolitinib improved the replication and oncolytic effect of VSV and produced inhibition of the drug-resistant pancreatic cancer cell line HPAF-II cells [115]. In the phase 3 clinical trial, ruxolitinib in combination with capecitabine was well tolerated in the treatment of refractory pancreatic cancer but did not improve survival [116]. However, Ruxolitinib may improve systemic inflammation in patients with metastatic pancreatic cancer, and in a randomized controlled trial that included 127 patients with metastatic pancreatic cancer who had failed treatment with gemcitabine, ruxolitinib significantly improved patient OS in a subgroup with serum C-reactive protein levels above 13 mg/L (hazard ratio = 0.47, 95% CI. 0.26~0.85, *p* = 0.011) [117]

AZD1480 is a JAK2 inhibitor that produces inhibition of a variety of tumors including diffuse large B-cell lymphoma, acute lymphoblastic leukemia, and lung cancer [118,119,120,121,122]. The clinical trial also evaluates the effectiveness of AZD1480 [118]. However, adverse events with AZD1480 occurred mainly neurologically, including dizziness, anxiety, ataxia, memory loss, hallucinations, and behavioral changes, and the cause of such adverse events is not known, leading to further clinical trials of the drug. In addition, JAK inhibitors such as momelotinib and ruxolitinib have also shown some adverse effects [123]. In patients with pancreatic cancer, 88% of patients reported grade 3 or 4 adverse events after using momelotinib [124]. As JAK is also involved in a variety of cellular activities, the use of JAK inhibitors can interfere with many normal physical and physiological processes in the body, such as myelosuppression, electrolyte disturbances, and viral reactivation.

#### 4.2.3. Other Indirect Inhibitors

Epidermal growth factor receptor tyrosine kinase inhibitors (EGFR-TKI), a representative of immune-targeted therapy for tumors, significantly prolong patient survival. In addition to its use in non-small cell lung cancer, it can also exert some anti-tumor effects on different types of tumors such as breast cancer and pancreatic cancer. When EGFR activity in cancer cells is inhibited, phosphorylation of STAT3 is diminished, thereby inhibiting cancer cell proliferation and improving resistance to the drug [125]. Erlotinib, an effective EGFR-TKI, significantly improved overall survival when combined with gemcitabine in patients with unresectable pancreatic ductal adenocarcinoma [126]. Erlotinib may become a stable targeted therapeutic drug for pancreatic cancer in the future due to its role in the EGFR/STAT3 pathway.

**Table 1 biomolecules-12-01450-t001:** Summary of effects of STAT3 target-related inhibitors in pancreatic cancer.

Targets	Inhibitors	Effects	Ref.
IL-6/IL-6R	Tocilizumab	Inhibition of stromal cell function and inhibition of EMT	[97,98,99]
	Siltuximab	Inhibition of cancer cell growth	[104]
	Bazedoxifene	Inhibition of cancer cell growth	[106,107,108]
JAK	Baricitinib	Inhibition of chemokines	[112]
	Ruxolitinib	Enhancing anti-tumor immunity and inhibiting inflammation	[114,117]
EGFR	Erlotinib	Inhibition of cancer cell growth	[126]
STAT3	C188-9	Reversing the hypermethylation status of tumor suppressor genes	[79]
	Shikonin	Promoting apoptosis of cancer cells	[83]
	Triptomycin A	Inhibiting the proliferation, migration and invasion of cancer cells	[84]

SRC, a representative member of the non-receptor tyrosine kinase family, plays a key role in the regulation of various signal pathways [127]. Aspergillus niger inhibits osteosarcoma cell growth and induces apoptosis by inhibiting the SRC/STAT3 signaling pathway in osteosarcoma [128]. In melanoma, inhibition of the SRC/STAT3 pathway reduces the expression of VEGF and matrix metalloproteinase-9, resulting in an anti-angiogenic effect [129]. As a botanical drug, oxalidaceae is used in the treatment of various diseases, and in pancreatic cancer cells can inhibit activated SRC, thereby affecting nuclear translocation including STAT3, resulting in low expression of the pro-angiogenic, apoptotic, and cell cycle vascular genes of STAT3 and exerting anti-cancer effects [130]. Some other indirect targets, such as VEGF, VEGF receptor, fibroblast growth factor receptor, and platelet-derived growth factor receptor are evaluated in currently on-going clinical trials. Current STAT3 inhibitors in clinical trials for cancer diseases are shown in Table 2.

### 4.3. Combined STAT3 Inhibitor and Immune Checkpoint Inhibitor (ICI)

Up-regulation of immune checkpoint molecules, including programmed cell death protein 1 (PD-1) and its ligand (PD-L1), has been shown to contribute to tumor immune escape [131]. In recent years, ICI has made progress in the treatment of cancer, but ICI is only effective in a subset of patients. JAK/STAT3 pathway can regulate the expression of PD-L1 and PD-L2 [132]. The TME of pancreatic cancer is usually immunosuppressed, so ICI is limited in the treatment of pancreatic cancer patients. Combination of targeted therapy and immunotherapy has been used in pancreatic cancer. Ruxolitinib combined with PD-L1 inhibitor (RMP1-14) can up-regulate the infiltration and activation of cytotoxic T lymphocyte, so as to overcome the resistance of pancreatic cancer to anti-PD-1 immunotherapy [114]. In addition, the combination of STAT3 indirect inhibitor (BP-5875) and anti-PD-L1 significantly inhibited the growth of pancreatic ductal adenocarcinoma [133]. The combination of STAT3 and ICI provides a promising strategy to improve the efficacy of current immunotherapy for pancreatic cancer.

## 5. Summary and Outlook

STAT3 is over-activated in pancreatic cancer and inhibitors targeting STAT3 have anti-tumor effects and also reverse the immune escape mechanism in TME. Therefore, STAT3 inhibitors may benefit pancreatic cancer patients by inhibiting tumor growth and mediating anti-tumor immunity.

Some STAT3 inhibitors have not shown a clear advantage in patients with pancreatic cancer [123,124,134]. Research on STAT3 inhibitors should focus on the following points: First, although STAT3 inhibitors have achieved good results in the experimental phase currently, they are limited by the limitations of monogenic drugs and further studies could be conducted to explore their use in combination with other classical chemotherapeutic drugs to improve the single- therapy of limitations and drug resistance to monotherapy. Second, indirect inhibitors of STAT3 have demonstrated some side effects in phase I/II clinical trials, such as neurological complications and bone marrow suppression, etc. Due to the low immunity and poor nutritional status of cancer patients, there is a need to further evaluate the safety of JAK and IL-6 inhibitors. In addition, the special genomic, proteomic, and metabolomic features of pancreatic cancer patients should also continue to be studied in depth, with a multi-omics macro-cut into the role of STAT3 in pancreatic cancer, leading to the design of more effective STAT3 inhibitors.

The concept of precision medicine guides us to develop individualized treatments for different patients. The study of STAT3 function and the use of STAT3 inhibitors will also lead to the study of the mechanism of pancreatic cancer and the development of drugs toward the path of precision medicine, thus prolonging the survival of patients.

## Figures and Tables

**Figure 1 biomolecules-12-01450-f001:**
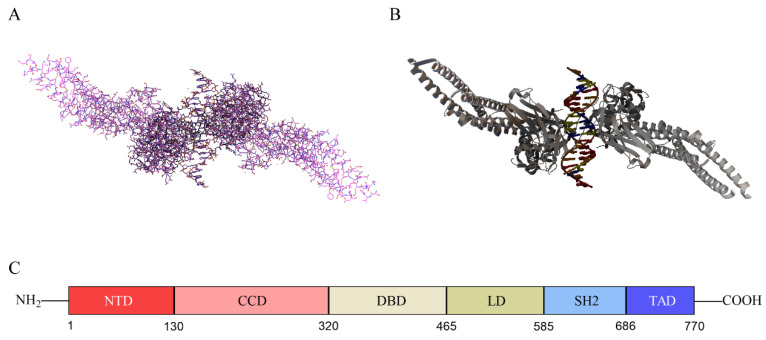
The three-dimensional structure and linear structure of STAT3. (**A**) The protein primary structure of STAT3; (**B**) the protein secondary structure of STAT3; (**C**) the linear structure of STAT3, including six structural domains, N-terminal domain (NTD), coiled-coil domain (CCD), DNA-binding domain (DBD), linking domain (LD), SH2 domain and transcriptional activation domain (TAD). STAT3 structure was retrieved from PDB database (PDB ID: 6QHD), then visualized with PyMol software.

**Figure 2 biomolecules-12-01450-f002:**
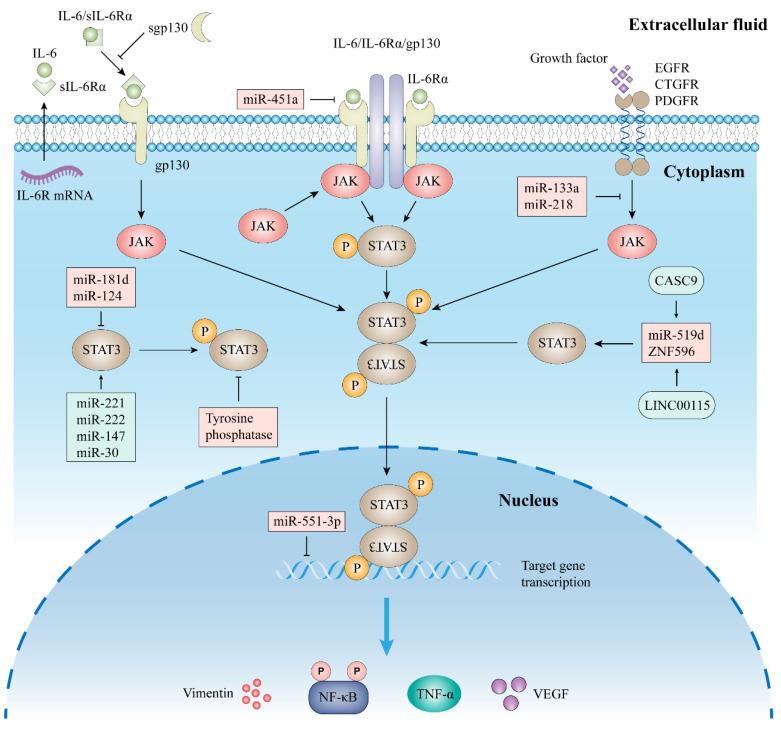
IL-6/JAK/STAT3 signaling pathway. IL-6 binds to the IL-6 receptor α (IL-6Rα), thereby inducing the formation of a complex consisting of two molecules each of IL-6, IL-6R, and GP130. Formation of this complex leads to activation of the JAK/STAT3 signaling pathway, resulting in the transcription of STAT3 target genes. Some growth factors can also activate the JAK/STAT3 pathway. Soluble IL-6R (sIL-6R) can be generated by alternative splicing of IL6R mRNA. sIL-6R binds to IL-6 to form a complex that binds and induces dimerization of GP130, leading to activation of downstream signaling pathways. Once bound to gp130, STAT3 is phosphorylated by JAK at tyrosine 705, leading to STAT3 dimerization and nuclear translocation, followed by STAT3-mediated transcription of target genes. Some molecules, such as miR-181d, miR-124, miR-451a, miR-551-3p, and tyrosine phosphatase can inhibit the STAT3 pathway. MiR-211, miR-222, miR-147, and miR-30 can promote STAT3 gene expression. When STAT3 is activated, it can promote the function of downstream molecules. Abbreviations: miR, microRNA; EGFR, epidermal growth factor receptor; CTGFR, connective tissue growth factor receptor; PDGFR, platelet-derived growth factor receptor.

**Table 2 biomolecules-12-01450-t002:** STAT3 inhibitors in clinical trials.

STAT3 Inhibitors	Cancer Types	Targets	Phase	NCT Number
IMX-110	Solid Tumor *	STAT3/NF-kB	I/II	NCT03382340
AZD9150	PC, NSCLC, CRC	STAT3	II	NCT02983578
Napabucasin	PC	STAT3	III	NCT02993731
Bazedoxifene	PC	IL-6 and gp130	NA	NCT04812808
Siltuximab	PC	IL-6	I/II	NCT04191421
CNTO 328	Solid Tumor	IL-6	II	NCT00841191
Ruxolitinib	PC	JAK1 and JAK2	I	NCT05440942
	PC	JAK1 and JAK2	II	NCT01423604
	PC, CRC	JAK1 and JAK2	I	NCT04303403
Itacitinib	Solid Tumor *	JAK1	I	NCT02646748
Ponatinib	CML	FGFR	II	NCT04043676
Sunitinib	RCCC	VEGFR, PDGFR	II	NCT03066427
	pNET	VEGFR, PDGFR	II	NCT02713763

Abbreviations: NA, not available; PC, pancreatic cancer; CRC, colorectal cancer; NSCLC, Non-small cell lung cancer; CML, chronic myeloid leukocytes; RCCC, renal clear cell carcinoma; pNET, pancreatic neuroendocrine tumor; FGFR, fibroblast growth factor receptor; VEGFR, vascular endothelial growth factor receptor; PDGFR, platelet-derived growth factor receptor. * Including pancreatic cancer.

## Data Availability

Not applicable.

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
