# Peer review of "STAT3 Inhibitors: A Novel Insight for Anticancer Therapy of Pancreatic Cancer"

_biomolecules, 2022, doi:10.3390/biom12101450_

Round 1

Reviewer 1 Report

Thanks to the authors, the manuscript is well written and organized. However; I have some comment as below: 

1- STAT3β is depicted, would of great help in understanding the favorable concept.

2- Figure 2 is a great one, but only has a title. It will definitely need description in the legend.

3- Although the medicines present in ongoing trials are somehow mentioned in the manuscript, a separate table in which STAT3 direct and indirect inhibitors present in clinical trials are introduced, could be beneficial. In this table can be categorized according to the type of cancer. Definitely, in this manuscript our interest is the molecules targeting pancreatic cancer or solid tumors.  

4- As targeting JAK/STAT can downregulate PD-1 and PD-L1 expression, STAT3 inhibitors can potentiate the effects of pembrolizumab and nivolumab. So, maybe a small paragraph about such combination could be advantageous

Reviewer 2 Report

The submitted manuscript should thoroughly review the methods to utilize STAT3 inhibitors in pancreatic cancer therapy. Below are listed significant comments regarding this manuscript.

1.       Section 2.1. The authors briefly introduced the STAT3 family and provided STAT1 and STAT3 as examples of the family. STAT1 is not the focus of this manuscript, and information that pertains to it should be removed.

2.       Section 2.1. The authors listed abbreviated names of domains of STAT3 without any explanation regarding their meaning and function. This is the essential information and should be at least briefly addressed.

3.       Section 2.1. STAT3 can be phosphorylated at Tyr 705 and/or Ser 727. Please add this information.

4.       Figure 1. Please describe the figures included in sections A, B, and C. In addition, please provide the origin of the sequence used to generate these figures and the software used for analysis and figure preparation.

5.       Section 2.2. There is a problem with the flow of this section. The paragraph focused on IL-6 should be moved after the first sentence of this section.

6.       In the fourth paragraph of section 2.2, it is tyrosine 705 that is phosphorylated, not serine (phosphorylated serine is in position 727).

7.       Figure 2. The figure legend providing an in-depth explanation should be included.

8.       Section 2.3 provides examples of STAT3 function from multiple tissues but does not provide good insight into the STAT3 pathway. Therefore, I would suggest removing this section entirely. Instead, I suggest providing a comprehensive review regarding STAT3 and its role in pancreatic cancer development, progression, metastasis, immunosuppression, and drug resistance. This section exists, but it is highly abbreviated and does not provide a complete picture of STAT3 role in PDAC. In addition, the authors should briefly describe the role of STAT3 in cancer epithelial cells, immune cells, and different stromal cells.

9.       The section regarding inhibitors is incomplete as well. There is a multiplicity of STAT3 pathway inhibitors that are not mentioned (examples of which are presented in the following reviews 10.1371/journal.pone.0252397 or /10.1186/s12943-020-01258-7).

Reviewer 3 Report

The review by Li, et al. offers a summative description of the role played by members of STAT protein family in cancer, specifically elucidating multiple connections between STAT pathway and pancreatic malignancies and provides an in-depth survey of currently available pharmacologic and molecular options to inhibit pathogenic STAT signaling and cancer.

The manuscript follows a well-organized flow of information, is written (with very few minor exceptions) by easy to follow language and will be undoubtedly of interest to read by a broad readership equally with research or clinical interests in mechanisms of pancreatic carcinogenesis and therapeutic strategies to interfere with them.

Of specific note, vast majority of cited references are dated in the last 2-3 years (covering 2020-2022 span in publication date) what further emphasize the originality aspect of the manuscript and rendering the review clearly distinct from numerous previously published on this topic summaries.

Several minor comments/recommendations to the text:

Line 59 – suggested “… can be phosphorylated for functional activation”

Line 115 – ADAM10 is an abbreviation for “A Disintegrin And Metalloproteinase Domain-containing protein 10”, needs to be corrected

Lines 151 and 158 – no need to write ’pericytes’ from capital letter, also consider using plural.

Line 325 – use period (not semicolon) to separate two sentences

Lines 349 and 350 – RA abbreviation not clarified in the text (evidently RA = rheumatoid arthritis)

Lines 445 – 450 – a long rather awkward sentence that is difficult to follow, consider revision.      
